# Genome-Wide Identification and Analysis of Stress Response of Trehalose-6-Phosphate Synthase and Trehalose-6-Phosphate Phosphatase Genes in Quinoa

**DOI:** 10.3390/ijms24086950

**Published:** 2023-04-09

**Authors:** Xiaoting Wang, Mingyu Wang, Yongshun Huang, Peng Zhu, Guangtao Qian, Yiming Zhang, Yuqi Liu, Jingwen Zhou, Lixin Li

**Affiliations:** Key Laboratory of Saline-Alkali Vegetation Ecology Restoration, Ministry of Education, College of Life Sciences, Northeast Forestry University, Harbin 150040, China

**Keywords:** trehalose biosynthesis, saline-alkali stress, quinoa, *CqTPS* and *CqTPP* gene families, transcriptomics and metabolomics

## Abstract

Saline-alkali stress seriously affects the yield and quality of crops, threatening food security and ecological security. Improving saline-alkali land and increasing effective cultivated land are conducive to sustainable agricultural development. Trehalose, a nonreducing disaccharide, is closely related to plant growth and development and stress response. Trehalose 6-phosphate synthase (TPS) and trehalose-6-phosphate phosphatase (TPP) are key enzymes catalyzing trehalose biosynthesis. To elucidate the effects of long-term saline-alkali stress on trehalose synthesis and metabolism, we conducted an integrated transcriptome and metabolome analysis. As a result, 13 *TPS* and 11 *TPP* genes were identified in quinoa (*Chenopodium quinoa* Willd.) and were named *CqTPS1-13* and *CqTPP1-11* according to the order of their Gene IDs. Through phylogenetic analysis, the CqTPS family is divided into two classes, and the CqTPP family is divided into three classes. Analyses of physicochemical properties, gene structures, conservative domains and motifs in the proteins, and cis-regulatory elements, as well as evolutionary relationships, indicate that the TPS and TPP family characteristics are highly conserved in quinoa. Transcriptome and metabolome analyses of the sucrose and starch metabolism pathway in leaves undergoing saline-alkali stress indicate that *CqTPP* and Class II *CqTPS* genes are involved in the stress response. Moreover, the accumulation of some metabolites and the expression of many regulatory genes in the trehalose biosynthesis pathway changed significantly, suggesting the metabolic process is important for the saline-alkali stress response in quinoa.

## 1. Introduction

The increasing soil saline-alkalization affects ecological environments and seriously reduces crop yields. On the contrary, the growing population in the world would require greater food production. This will lead to a contradiction between the supply and demand of food in the near future. The improvement of saline-alkali land can increase the area of the effectively cultivated land, which is very important for the sustainable development of agriculture. The total area of saline-alkali land in China is about 9.91 × 10^4^ km^2^ [1]. The saline-alkali land in northeast China, also known as soda saline-alkali land, contains a high concentration of carbonate, including Na_2_CO_3_ and NaHCO_3_, etc. [2,3].

Salt stress leads to ion toxicity and osmotic stress, disrupting metabolism and ion homeostasis, resulting in impaired photosynthesis, disordered metabolic process, and retarded root development, and thus inhibits plant growth and development which leads to a decline in crop yields [4,5,6,7]. In addition to ion toxicity and osmotic stress, alkali stress aggravates the damage to cell structure and activities. Saline-alkali stress causes serious interference with metabolic processes (e.g., carbohydrate degradation and nitrogen metabolism retardation) and significant changes in the contents of inorganic and organic ions, carbohydrates (amino acids and sugars, e.g., proline, sucrose, trehalose, and fructose), ROS, and MDA, etc. [8,9,10].

The carbohydrates are mainly synthesized by starch and sucrose metabolism and related pathways, e.g., glycolysis/gluconeogenesis, and amino sugar and nucleotide sugar metabolism [11]. The starch and sucrose metabolic pathway plays important roles in plant growth and development, yield quality, and stress response. The products and intermediate products not only provide carbon sources for plant growth and development but also participate in many biological processes, acting as signal molecules [12]. Various enzymes, e.g., sucrose phosphate synthase (SPS), sucrose synthase (SUS), and invertase (INV), participate in sucrose biosynthesis, distribution, and accumulation in plants [13]. The content of sucrose is an important factor for controlling the transport tempo of nutrients, directly affecting the growth, yield, and quality of crops [14]. In plants, starch is the most abundant carbohydrate reserve as the source of carbon and energy. Glucose is the precursor of starch biosynthesis in the chloroplasts and amyloplasts [15,16]. Starch metabolism is a complex and tightly regulated process involving several classes of enzymes, including ADP glucose phosphorylase (AGPase), starch synthase (SS), starch branching enzyme (SBE), starch debranching enzyme (DBE), phosphorylase (PHO), and disproportionation enzyme (DPE).

Trehalose (α-d-glucopyranosyl (1-1)-α-d-glucopyranoside), a nonreducing disaccharide, is produced in the starch and sucrose metabolism process [17]. Trehalose affects carbon allocation in plants and plays an important role in plant growth and development and stress response. Trehalose is synthesized from UDP-glucose (UDPG) and glucose 6-phosphate (Glc6P), which are catalyzed by trehalose-6-phosphate synthase (TPS) to form trehalose-6-phosphate (T6P), and then trehalose-6-phosphate phosphatase (TPP) dephosphorylates T6P to produce trehalose [17]. In addition to acting as a precursor of trehalose, T6P is also an important signal molecule for plant growth and development via dynamically regulating sucrose metabolism [18,19]. T6P inhibits the activity of sucrose nonfermenting-1-related protein kinase 1 (SnRK1) in plants [20]. The interaction between T6P and SnRK1 facilitates the maintenance of sugar homeostasis in plants [21].

In addition to catalyzing trehalose biosynthesis, TPSs and TPPs can improve plant stress tolerance. Salt stress induces *TPS* expression in wheat and improves *OsTPP1* expression in rice [22,23]. Overexpression of *OsTPS1* in rice increases the aglucon and proline content and promotes the upregulation of some stress-response genes, thereby enhancing tolerance to cold, drought, and salinity in rice [23,24]. Ectopic overexpression of the yeast *TPS1* gene in potato also enhances drought tolerance [25]. The *Arabidopsis TPS1* gene plays a key role in embryo development and flowering, and *AtTPS1* depletion mutants are embryonically lethal [26]. Although the TPS/TPP pathway has been well-characterized in many species, little is known about the *TPPs* and *TPSs* in quinoa.

Quinoa (*Chenopodium quinoa* Willd.) is an annual dicotyledonous herbaceous crop of the *Amaranthaceae* family. Quinoa grains have high protein contents and excellent amino acid composition, including all essential amino acids required for the human body. Quinoa grains contain abundant natural antioxidant substances, vitamins, dietary fiber, and minerals [27]. Moreover, quinoa is cholesterol- and gluten-free and is an ideal food for certain patients or sensitive people. Based on these characteristics, the research on quinoa has received extensive attention. Moreover, the publication and upgrading of the quinoa genome sequence greatly promoted the research on quinoa [28].

In this study, we identified 13 *TPS* genes and 11 *TPP* genes in quinoa by genome sequence search and conserved domain search. The *CqTPS* and *CqTPP* genes were classified according to conservative domain and motif analysis. Furthermore, transcriptome and metabolome analyses were performed and clarified the important roles of *CqTPS* and *CqTPP* genes as well as starch and sucrose metabolism in the saline-alkali stress response in quinoa. Our study provides basic information for understanding the characteristics of the *CqTPS* and *CqTPP* gene families in quinoa and their functions in response to saline-alkali stress.

## 2. Results

### 2.1. Genome-Wide Identification of TPS and TPP Gene Family in Quinoa

In order to identify TPS and TPP family members in quinoa, the amino acid sequences of AtTPS1 (Accession No.: XM_002889154.1) and AtTPP1 (Accession No.: NM_001344938.1) were used as references to perform local BLAST based on the genome database of quinoa (*Chenopodium quinoa* Willd.). The family members were further confirmed according to the conservative domains. As a result, thirteen TPS proteins and eleven TPP proteins were identified in quinoa (Table 1 and Table 2) and were named CqTPS1-CqTPS13 and CqTPP1-CqTPP11 according to the Gene IDs. The CqTPS protein length is between 501 (CqTPS6) and 1173 (CqTPS12) amino acid residue numbers (aa), but most of the CqTPSs contained between 830 and 1173. This might be associated with differences in the protein domain. And the molecular weight of CqTPSs is between 56.16 kDa (CqTPS12) and 130.87 kDa (CqTPS12). The isoelectric point (pI) of all CqTPS proteins is less than 7; namely, they belong to acidic proteins. The CqTPP protein length is between 324 (CqTPP9) and 387 (CqTPP6) aa, and the molecular weight is between 37.08 kDa (CqTPP9) and 43.28 kDa (CqTPP6). Most CqTPPs belong to basic amino acids with an IP higher than 7, apart from TPP9 and TPP11, the two acidic proteins with pI5.97 and 6.80, respectively.

### 2.2. Phylogenetic Analysis of TPS and TPP Family Genes in Quinoa

To assess the TPS or TPP evolutionary relationship of *Chenopodium quinoa* Willd, *Arabidopsis thaliana* (L.), and *Gossypium hirsutum Linn*. [29,30,31,32], the phylogenetic trees of the TPS and TPP gene families were constructed, respectively. According to the phylogenetic tree, the *TPS* family members were divided into two classes (Figure 1A). Class I includes four *AtTPSs*, six *GrTPSs*, and six *CqTPSs*, and Class II includes seven *AtTPSs*, nine *GrTPSs*, and seven *CqTPSs*. The TPP family is divided into three classes (Figure 1B). Class I includes seven *AtTPPs*, six *GrTPPs*, and four *CqTPPs*, Class II includes three *AtTPPs*, four *GrTPPs*, and four *CqTPPs*, and Class III only includes three *CqTPPs* and two *GrTPPs* without an *AtTPP* homologue.

### 2.3. Analysis of Primary Structures of Genes and Proteins of CqTPSs and CqTPPs

The conservative structure analysis indicates that the TPS domain (Pfam: PF00982) is unique to the CqTPS family proteins, and the TPP domain (Pfam: PF02358) is shared by CqTPS and CqTPP proteins (Figure 2A,B). Class I CqTPS enzymes contain a catalytic triad of residues, Arg (R)/Lys (K)/Glu (E), that is required for T6P synthesis. Moreover, Class I CqTPSs have a SUMOylation site at the C-terminal. Class II CqTPSs have an incomplete triad lacking ‘R’ residue in TPS domain, and no sumoylation site at the C-terminal (Figure 2A, Appendix A). Then, we used MEME online tools (https://meme-suite.org/meme/doc/meme.html, accessed on 16 December 2022.) to analyze the motif distribution in CqTPS and CqTPP proteins. Ten conservative motifs were identified in CqTPS proteins, with a length of 21–50 aa (Figure 2C), and their sequence information is shown in Appendix A. Most CqTPS proteins contain all ten motifs, except for CqTPS7 lacking motif 10, TPS6 lacking motifs 1, 4, 6, and 5, and resulting in shorter proteins. According to the annotation, motifs 4, 6, 1, 5, 8, and 3 constitute the TPS domain, and motifs 10, 2, 9, and 7 constitute the TPP domain. The gene structure analysis based on GFF3 annotation indicates that the number of introns has a significant difference between the two classes. Class II *CqTPS* genes have only two introns. However, Class I *CqTPS* genes have a large number of introns, e.g., *TPS6* has eight introns, and *TPS5, 9, 10, 11,* and *12* genes have ≥14 introns (Figure 2E).

The motif distribution analysis identified ten conservative motifs in CqTPP proteins (Figure 2D). The length of these conservative motifs ranged from 15 to 50 aa, and their sequence information is shown in Appendix A. The motif comparison indicates that Class I and Class II CqTPP proteins share some common motifs. All CqTPPs contain motifs 4, 1, 3, 7, 2, and 5. Class I and Class II CqTPPs also have motifs 8 and 6 at the N terminal, while Class II CqTPPs have a motif 9 in the middle of the proteins. In Class III, CqTPP5 has no additional motif, but CqTPP4 and 9 have a motif 10 inserted after motif 1 (Figure 2F). There is no significant difference in the gene structure of *CqTPPs*. The above results indicate that *CqTPS* and *CqTPP* genes have undergone functional differentiation during evolution.

### 2.4. Analysis of Cis-Acting Elements in the Promoters of CqTPS and CqTPP Genes

The sequences 2000 bp upstream of the start codon were selected as the promoters for cis-regulatory elements analysis. A total of 50 elements were identified in the promoters of *CqTPSs* and *CqTPPs* (Figure 3A,B). The *cis*-acting elements are classified into five categories, including Light Response, Plant Hormone, Plant Growth, Stress Response and Other. The composition of *cis*-acting elements in the *CqTPS* and *CqTPP* families is similar. The Light Response group contains the most elements, accounting for more than 60% of all elements. Among the Light Response elements, the number of Box 4 and G-box are the largest, followed by the GT1-motif element. It has been demonstrated that G-box in salt mustard participates in the salt stress response [33], and GT1-motif participates in nitrogen regulation in soybean [34]. In the Plant Hormone group, the number of ABREs, CGTCA-motifs, and TGACG-motifs are the largest. ABRE is demonstrated to be involved in the abscisic acid response, and the TGACG-motif participates in the methyl jasmonate response, indicating that the trehalose synthesis in quinoa is modulated by abscisic acid and jasmonate, which are stress responsive in plants [35]. Among Plant Growth elements, the number of AREs and Circadians are the largest. ARE is required for anaerobic induction, and Circadian element regulates circadian rhythm. Among the Stress Response elements, MBSs and TC-rich repeats are the most abundant. MBS is the MYB binding site involved in drought induction, and TC-rich repeats are involved in the defense and stress responses [36]. These results indicate that *CqTPS* and *CqTPP* genes are widely involved in various physiological and biochemical activities in plants, and responses to light, phytohormones, and abiotic stress.

### 2.5. Transcriptome and Metabolome Analyses of Trehalose Biosynthesis in Quinoa Leaves under Saline-Alkali Stress

To investigate the saline-alkali stress response of CqTPSs and CqTPPs and related metabolites such as trehalose, glucose, and sucrose, etc., we performed integrated transcriptome and metabolome analysis using quinoa leaves undergoing carbonate treatment. Two weeks after the treatment, the quinoa plant height reduced significantly compared with that of untreated plants (Appendix A) and the total chlorophyll content declined significantly (Appendix A), indicating the saline-alkali condition seriously affected quinoa growth and development. Transcriptome analyses of *CqTPS* and *CqTPP* genes indicate that *CqTPS4* was significantly upregulated [log_2_ fold change (FC) = 1.77], while *CqTPS3* (log_2_FC = −1.40), *CqTPS8* (log_2_FC = −1.25), and *CqTPP4* (log_2_FC = −1.82) were significantly downregulated, and expression of the other genes changed but not significantly (Figure 4A,B). GO enrichment analysis indicates that the trehalose-related processes such as trehalose-phosphatase activity, the trehalose metabolic process, and the trehalose biosynthesis process were well enriched (Figure 4D, arrows). On the other hand, the contents of some metabolites in the starch and sucrose metabolism pathway were altered significantly. For example, UDP-Glucose (log_2_FC = −12.80), Glucose-1-phosphate (log_2_FC = −1.61), N-Acetyl-D-glucosamine (log_2_FC = −1.61), D-Melezitose (log_2_FC = −1.31), D-Fructose 6-Phosphate (log_2_FC = −12.80), 3-Dehydro-L-Threonic Acid (log_2_FC = −1.21), and D-Glucose 6-phosphate (log_2_FC = −1.28) decreased significantly, but D-Glucose (log_2_FC = −1.01) decreased slightly, while D-Panthenol (log_2_FC = 1.15) increased significantly (Figure 4C). These results indicate that the energy metabolism was severely inhibited under saline-alkali stress, and the CqTPSs and CqTPPs as well as trehalose metabolism are important for the saline-alkali stress response in quinoa.

Since the TPS- and TPP-regulated trehalose biosynthesis processes are in the starch and sucrose metabolism pathway, GO analysis of this pathway was conducted (Figure 5A, Appendix A). Among the metabolites, UDP-Glucose (UDPG) content changed the most (log_2_FC = −12.80), and D-Glucose-1P, D-Glucose-6P, and D-Glucose, etc., also reduced. UDPG is mainly synthesized from Glucose-1P catalyzed by UGP2, or from sucrose catalyzed by sucrose synthases (SUSs) [37], the expression of one *SUS* increased and two declined. TPSs catalyze the synthesis of T6P from UDPG and D-Glucose-6P, and then TPPs dephosphorylate T6P to generate trehalose [38]. The expression of one *CqTPS* was upregulated and two *CqTPSs* and one *CqTPP* were downregulated. Trehalose and glucose/maltose can be converted into each other, and maltose can be converted into glucose catalyzed by maltodextrin glucosidase (malZ) [39], while the expression of *malZ* increased. In order to verify the transcriptome data, we performed an RT-qPCR analysis to check the expression of the related DEGs, and the results were basically consistent with the transcriptome data (Figure 5B). These results prove that CqTPSs and CqTPPs and the starch and sucrose metabolism pathway play important roles in the carbonate-induced saline-alkali stress response in quinoa.

### 2.6. Analysis of CqTPS Genes’ Response to Saline-Alkali Stress

To characterize the regulatory roles of *CqTPS* genes in response to saline-alkali stress, we determined the growth of yeast (*Saccharomyces cerevisiae*) harboring *CqTPS4/8* genes exposed to saline-alkali stress. The results indicate that *CqTPS4*- and *CqTPS8*-expressing yeast cells exhibited an increased survival ratio compared to the control (with empty vector) on an SC-Ura medium containing 25 mM NaHCO_3_ with pH = 5, 7, 8, and 9 (Figure 6), suggesting that *CqTPS4* and *CqTPS8* are involved in the saline-alkali stress response.

## 3. Discussion

TPSs and TPPs are key enzymes for trehalose biosynthesis and are important for plant development and stress response. To date, *TPS* and *TPP* family genes have been identified in all major plant taxa and microbes such as bacteria, fungi, and yeast, etc. [40,41,42]. In this study, we identified 13 *TPS* genes and 11 *TPP* genes in the quinoa genome (*Chenopodium quinoa* Willd.). Compared with cotton and *Arabidopsis*, CqTPSs were divided into two classes, and CqTPPs were divided into three classes, according to their protein conservative domains. The Class III TPP group does not include *Arabidopsis* proteins, and the members lack N-terminal motifs 6 and 8, which exist in Class I and Class II TPPs (Figure 2D). This may be due to the gene variation resulting from exon/intron gain/loss, exonization/pseudoexonization, and/or insertion/deletion, which lead to functional differentiation during evolution [43,44].

The AtTPS1 protein has a catalytic triad of residues, Arg (R)/Lys (K)/Glu (E), which is critical to TPS activity, and contains a putative SUMOylation site in the TPP domain [45]. Lack of the catalytic triad results in loss of TPS activity. Class I AtTPS proteins with the catalytic triad residues in the TPS domain are demonstrated to have catalytic activity, except for AtTPS3, a possible pseudogene [46]. These structure characteristics are also conserved in quinoa. Among the 13 CqTPSs, Class I CqTPSs have the catalytic triad of residues in the TPS domain and a SUMOylation site at the C-terminal (Figure 2A) [45,46]. According to the conservative domains and motifs, it is speculated that Class I CqTPSs may have TPS activity, and, of course, this needs to be verified by experiments. The Class II CqTPSs have an incomplete triad lacking the ‘R’ residue in the TPS domain, and no sumoylation site is found at the C-terminal, suggesting they may not have TPS activity. These structural features are consistent with the *Arabidopsis* Class II members, which have been elucidated to have no TPS catalytic activity [47,48]. The TPP domain in AtTPS1 is likely crucial for plant development, although it lacks some of the residues in the active site of the TPP enzyme [42]. Interestingly, despite the high conservation of TPP activity sites, none of the class II AtTPSs have TPP activity [49,50,51]. Meanwhile, the TPP domain in Class I and Class II CqTPSs is highly similar to that of AtTPSs, implying that they are likely to have similar functions (Appendix A). Although the Class II TPS proteins may have no TPS and TPP enzyme activities, they are proposed to regulate Class I TPS enzyme activity based on the association of rice class I and II proteins [52], and act as signaling proteins based on their conservative ligand-binding sites [42]. There is much evidence implicating that Class II TPS proteins are involved in stress response. For instance, TPS5 is involved in ABA signaling [53], TPS5 in thermotolerance [54], TPS5 in basal pathogen defense [43], and TPS11 in cold tolerance [44] and aphid resistance [55]. In quinoa leaves, the expression levels of Class II genes, *CqTPS3*, *CqTPS4,* and *CqTPS8*, were altered significantly under saline-alkali conditions, suggesting that Class II genes in quinoa also respond to stress. On the other hand, all CqTPS proteins do not have a nuclear localization signal (NLS), suggesting they cannot enter the nucleus autonomously.

In addition to TPP catalytic activity, some *AtTPP* genes are associated with an abiotic stress response. For example, *AtTPPD* is responsible for salt and oxidative stress resistance [56], and *AtTPPF* and *AtTPPI* are involved in drought response [56,57]. Moreover, the rice *OsTPP1* and *OsTPP2* are involved in the cold stress response [58,59], and *OsTPP7* regulates anaerobiosis resistance during germination [60]. In quinoa leaves, expression of *CqTPP4* decreased significantly under saline-alkali stress (Figure 6B), suggesting it is involved in the stress response. The transcription levels of other *CqTPP* genes also changed under the same conditions but not significantly. Further study is needed to clarify the association of *CqTPP* genes with different stresses.

When plants are under stress, the related transcription factors (TFs) are activated to bind specific cis-acting elements to regulate the expression of downstream stress-response genes. In this study, many cis-acting elements related to stress (A-box, MBS, LTR, TC-rich repeats, and WUN-motif), phytohormones (ABRE, GCTGA-motif, etc.), and light (G-box, box4, etc.) were identified in promoters of *CqTPS* and *CqTPP* family genes. In quinoa leaves under saline-alkali stress, the expression levels of *CqTPS4*, *3*, *9,* and *CqTPP4* were altered significantly (Figure 4A,B). Among them, *CqTPS4*, the only upregulated gene, contains six GATA-motifs, three Box4s, two Gap-boxes, and two G-boxes in its promoter (Figure 3A), suggesting these elements may be closely related to the response to saline-alkali stress. The other three *CqTPS* and *CqTPP* genes were significantly downregulated under saline-alkali stress. This is also observed in previous studies, that show that some *TPS* or *TPP* genes are up- and some other genes are downregulated. This may be due to the complexity of regulatory mechanisms of trehalose biosynthesis as well as the functional division of these genes, e.g., some Class I *TPS* genes are responsible for plant growth and development, and some Class II *TPS* genes are related to the stress response. Another possible reason is the plant development stage and stress treatment duration. Two-week treatment is a long-term stress, the quinoa plants may have adapted to the condition and have adjusted the metabolic system and regulatory machineries. For example, the stabilized expression levels of *CqTPS* genes lead to a proper level of T6P, which subsequently stabilizes the abundance of sucrose and trehalose as well as related physiological activities.

Sucrose is the final product of the photosynthesis of green plants and functions as an important carbon source for cell activities. Carbohydrates from sucrose account for 90% of plant biomass. At the same time, sucrose and starch metabolism, which produces signal molecules such as glucose, fructose, and T6P, is one of the key pathways to regulate plant growth and development and abiotic stress response [61,62,63]. Trehalose and T6P are important intermediates in sucrose and starch metabolism, and trehalose biosynthesis is essential for plant growth and development [64]. In most flowering plants, the trehalose content is extremely low and is barely detectable [65,66]. Under stress conditions, trehalose functions as an osmoprotectant to protect the cellular structure and cell membrane from stress-induced damage [38] and improve plant tolerance to various abiotic stresses, especially salinity and drought stresses [36]. T6P links plant growth and development with the status of sucrose metabolism [67]. In *Arabidopsis* seedlings, the T6P content is highly correlated with the level of sucrose, and dynamically responds to the fluctuation of sucrose and gives feedback. Therefore, T6P is considered a key signaling metabolite, acting as a plant ‘insulin’ to regulate sucrose utilization. [68].

Surprisingly, despite being under saline-alkali treatment and despite the alteration of expression of *CqTPS* and *CqTPP* genes and significant decrease of contents of UDPG and D-Glucose-6P, the contents of sucrose, trehalose, and T6P in quinoa leaves did not change significantly, whereas the contents of UDPG and D-Glucose-6P, the substrates for T6P generation, decreased significantly (Figure 4C and Figure 5A). Similar phenomena were also observed in maize *ra3* mutants, the *TPP* loss-of-function alleles, and there were no significant differences in T6P or trehalose levels compared to wild-type [66]. Moreover, there were no significant differences in T6P levels between wild-type, *ra3*, and *ra3 tpp4* mutants [69]. Since T6P plays a central role in controlling growth and development, a modicum change in T6P level may cause serious effects on plants. Significant changes have also occurred in the transcription levels of many genes in starch and sucrose metabolic pathways, such as amylase genes, *SUS* (sucrose synthase), *glgA* (glycogen synthase), and *glgC* (ADPglucose pyrophosphorylase) genes, etc. (Figure 5), which consequently led to changes in metabolites. GO Enrichment analysis indicates that the trehalose-related processes such as trehalose-phosphatase activity, the trehalose metabolic process, and the trehalose biosynthesis process were well enriched in DEGs (Figure 4D, arrows). On the other hand, the contents of some metabolites in starch and sucrose metabolism pathway altered significantly. Therefore, there may be tight regulatory machineries to rescue the change of T6P, such as the complementation from other functional TPS and/or TPP genes, or other unknown pathway(s). In the case of this study, the two-week treatment belongs to a long-term stress, and the saline-alkali stress was moderate, thus the quinoa plants may have adapted to the stress and have adjusted the T6P level, and subsequently stabilized the contents of trehalose and sucrose. Quinoa is a species with strong resistance to stress; whether the adjustment of T6P is due to this characteristic of quinoa or the nature of many plants is unknown. In source leaves, much of the UDP-Glc is consumed by sucrose synthesis [70]; whereas in growing tissues, it is the direct substrate for the synthesis of cellulose and hemicelluloses, and a precursor for other nucleotide sugars that are needed for the synthesis of noncellulosic cell wall polysaccharides [71,72]. Therefore, the remarkable decrease of UDPG in leaves of developing quinoa plants may be due to consumption in sucrose synthesis and cell wall deposition, as well as in other metabolic pathways, such as amino sugar and nucleotide sugar metabolism (Figure 5A).

Moreover, the expression of many regulatory genes in the sucrose and starch metabolism pathway changed, including sucrose synthases (*SUSs*) and amylases, etc. (Figure 5A). SUSs catalyze the reversible conversion of sucrose and fructose and produce UDPG [73]. The abundance of SUSs affects the thickness of the secondary cell wall, sharevegetative growth, and mechanical strength [74]. On the other hand, the expression of *SUSs* is affected by salt stress [75]. The amylases include alpha-amylase (AMY) and beta-amylase (β-amylase). AMY is a crucial enzyme that functions throughout the whole life cycle of angiosperm [76]. Alteration of the expression of these genes suggests that the sucrose and starch metabolism pathway is crucial for the saline-alkali stress response.

## 4. Materials and Methods

### 4.1. Genome-Wide Identification of TPS and TPP Family Members in Quinoa (Chenopodium quinoa Willd.)

From Ensemble Plants (http://plants.ensembl.org/index.html, accessed on 5 December 2022.), we downloaded the genome file and GFF3 file of quinoa (http://plants.ensembl.org/index.html, accessed on 5 December 2022.) [77], from TAIR (https://www.Arabidopsis.org/, accessed on 5 December 2022.), we downloaded *Arabidopsis* TPS and TPP protein sequences.

### 4.2. Bioinformatics Analysis of Quinoa TPS and TPP Family Genes

#### 4.2.1. Phylogenetic Analysis

The phylogenetic trees of TPS and TPP families, containing quinoa, *Arabidopsis,* and cotton proteins, were constructed based on multiple sequence alignments using MAGA 11 software [78] and the maximum likelihood model (bootstrap is 1000). iTOL (http://itol.embl.de/, accessed on 12 December 2022.) website was used to optimize the evolutionary tree.

#### 4.2.2. The Physicochemical Properties, Conserved Motif Analysis

The protein length, isoelectric point (pI), and molecular weight (MW) were predicted using ExPASy website (https://www.expasy.org/, accessed on 8 December 2022.) [79].

The conserved motifs were retrieved by searching MEME website (https://meme-suite.org/meme/doc/meme.html, accessed on 16 December 2022.) [80]. The maximum retrieval value was set to 10, and the other parameters were default. InterProScan software (interproscan-5.55-88.0-64-bit) was used to annotate the motifs.

#### 4.2.3. Gene Structure and Cis-Acting Element Analysis

The gene structure of the *CqTPS* and *CqTPP* family genes was analyzed using TBtools combined with the GFF3 gene annotation data, and to plot the exon–intron diagram.

The 2kb sequences upstream of the start codons of *CqTPS* and *CqTPP* genes were screened using TBtools and used as promoter sequences for analysis. plantCARE [81] (https://bioinformatics.psb.ugent.be/webtools/plantcare/html/, accessed on 20 December 2022.) was used to investigate the cis-acting elements in promoters to predict the roles of genes in stress responses.

### 4.3. Plant Materials, Growth Conditions, and Stress Treatments

Quinoa (Jiaqi #3) plants grew under a 16 h light/8 h dark cycle at 22 °C. Two-week-old quinoa seedlings were treated with a solution containing 100 mM Na_2_CO_3_:NaHCO_3_ = 1:9 once every 5 days, three times in total. The control group was treated with water. The leaves were harvested randomly five days after the third treatment, then sent for RNA sequencing and an ultraperformance liquid chromatography–electrospray ionization–tandem mass spectrometry (UPLC–ESI–MS/MS) analysis.

### 4.4. RT-qPCR Validation

The total RNA used for RT-qPCR (reverse transcription quantitative PCR) was consistent with that for RNA sequencing. RT-qPCR was performed following the manufacturer’s instructions for Ultra SYBR Mixture (Low ROX) on the ABI7300 real-time PCR system (Applied Biosystems, Waltham, MA, USA). *UBQ9* (AUR62020068) was used as the reference gene for normalizing mRNA transcription [82]. The relative expression level was calculated by the 2^−△△CT^ method [83]. All RT-qPCR analyses were set with 3 technical repeats. The primers are listed in Appendix A.

### 4.5. Stress Tolerance Assays of CqTPS4 and CqTPS8 in Yeast

The cds of *CqTPS4* and *CqTPS8* were amplified from the quinoa cDNA library using specific primers (listed in Appendix A). The PCR products were inserted into the *pYES2* vector, and the recombined *pYES2-TPS4* and *pYES2-TPS8* plasmids were transformed into yeast (*Saccharomyces cerevisiae*) strain INVSC1, respectively. The *pYES2* empty vector was used as a control.

The transformed yeast was cultured overnight in SC-Ura liquid medium containing 2% glucose at 30 °C. After adjustment of OD600 to 0.4, the culture was diluted according to the indicated gradient on an SC-Ura medium containing 25 mM NaHCO_3_ with pH 5, 6, 7, 8, and 9, respectively, at 30 °C for 24 h.

## 5. Conclusions

In this study, we identified 13 *CqTPS* genes and 11 *CqTPP* genes in the quinoa genome and analyzed their physicochemical properties, gene structures, conservative domains and motifs in the proteins, and cis-regulatory elements, as well as evolutionary relationships. The results indicate that the TPS and TPP family characteristics are highly conserved in quinoa. Transcriptome and metabolome analyses of the sucrose and starch metabolism pathway in leaves indicate that *CqTPP4* and Class II *CqTPS4*, *3*, *9* genes, and trehalose biosynthesis are important for the saline-alkali stress response in quinoa. This study may help to explain the biological activities of CqTPS and CqTPP proteins in developmental processes and stress responses in quinoa. However, our knowledge of their precise biological role is still lacking. Thus, in order to provide important insights to help others for developing crop cultivars resistant to unfavorable stress conditions, an extensive functional validation study of CqTPSs andCqTPPs is necessary.

## Figures and Tables

**Figure 1 ijms-24-06950-f001:**
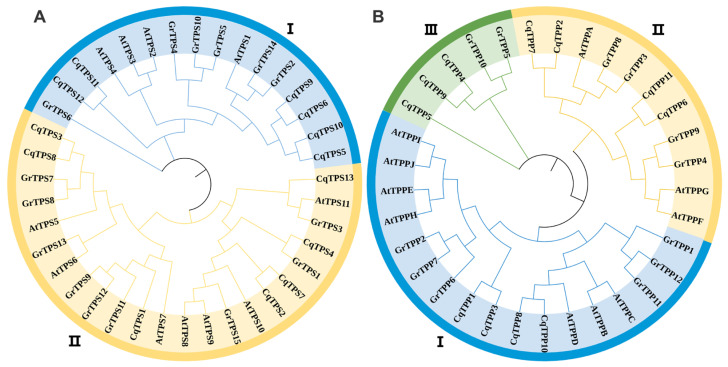
Phylogenetic analysis of *TPS* and *TPP* gene familes of *Chenopodium quinoa* Willd, *Arabidopsis thaliana* (L.), and *Gossypium hirsutum Linn*. (**A**) A phylogenetic tree containing 13 *CqTPSs*, 11 *AtTPSs*, and 15 *GrTPSs*, which are divided into two Classes distinguished in different colors. (**B**) A phylogenetic tree containing 11 *CqTPPs*, 10 *AtTPPs*, and 12 *GrTPPs*, which are divided into three Classes distinguished in different colors. I, II and III represent Class I, Class II and Class III respectively.

**Figure 2 ijms-24-06950-f002:**
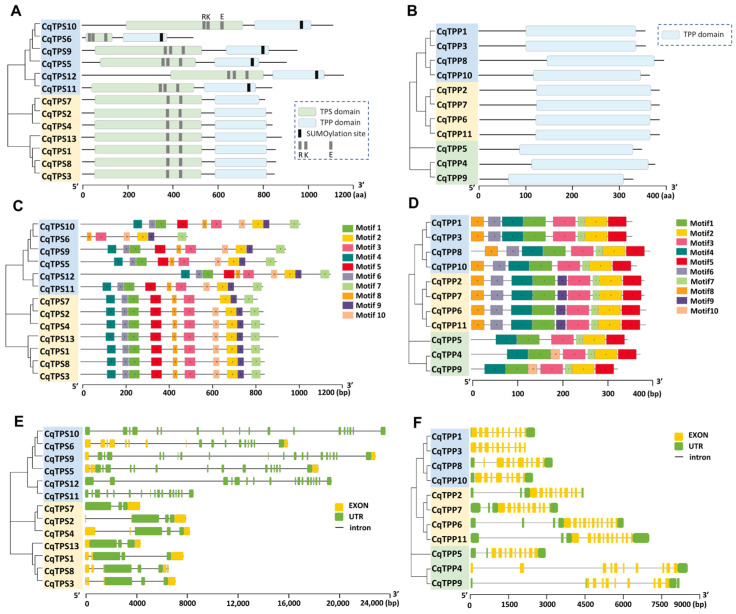
Analysis of gene and protein primary structures of CqTPSs and CqTPPs. (**A**,**B**) Conservative domain analysis of CqTPS (**A**) and CqTPP (**B**) proteins. Class I CqTPS proteins contain a catalytic triad residue, Arg (R)/Lys (K)/Glu (E) (dark gray bars), and a SUMOylation site at the C-terminal (black bars). (**C**,**D**) Motif distribution analysis of CqTPS (**C**) and CqTPP (**D**) proteins. (**E**,**F**) Gene structure analysis of *CqTPSs* (**E**) and *CqTPPs* (**F**).

**Figure 3 ijms-24-06950-f003:**
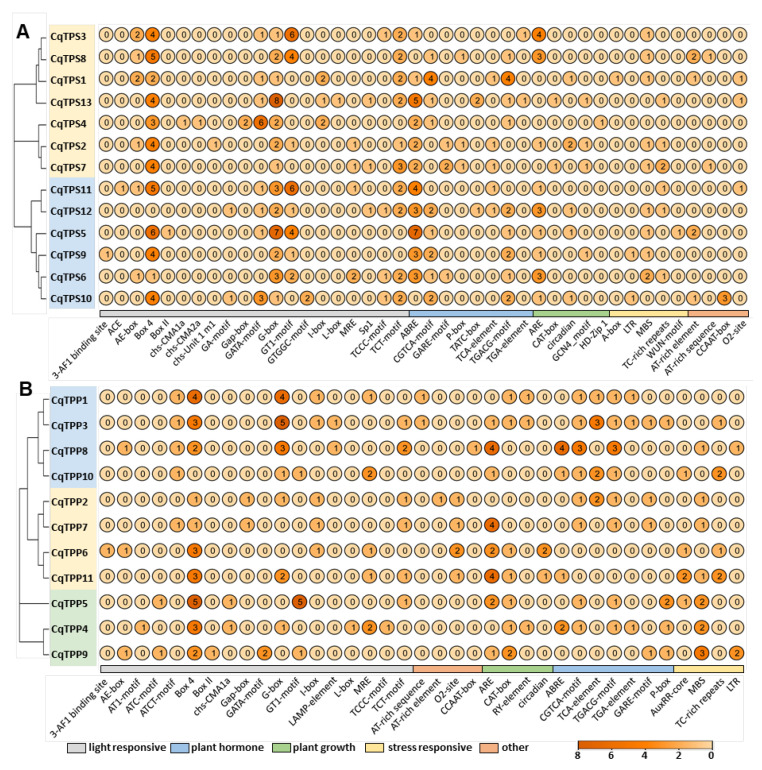
Analysis of cis-acting elements in promoters of *CqTPS* (**A**) and *CqTPP* (**B**) genes. The categorized groups are indicated by color bars. The number of each element is indicated in the circles and is also shown by different colors presented in a color scale in the lower panel.

**Figure 4 ijms-24-06950-f004:**
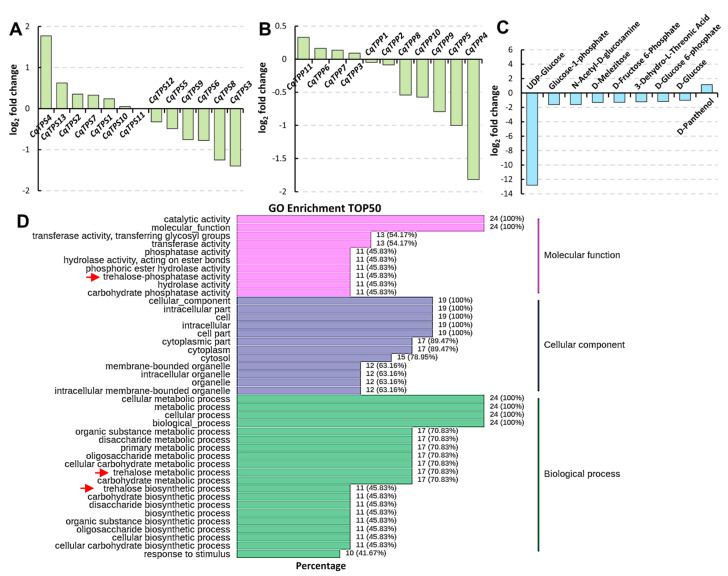
Transcriptome analysis of *CqTPSs* and *CqTPPs* and related processes and metabolism analysis of related metabolites in quinoa leaves undergoing saline-alkali stress. (**A**,**B**) Transcriptome analysis of expression levels of *CqTPS* and *CqTPP* family genes in quinoa leaves. Two-week-old quinoa seedlings were treated with a carbonate solution containing 100 mM Na_2_CO_3_:NaHCO_3_ = 1:9 for two weeks. (**C**) Metabolism analysis of the differently accumulated metabolites (DAMs) in the starch and sucrose metabolism pathway. (**D**) Go enrichment analysis of total differently expressed genes (DEGs). Arrows, trehalose-related processes.

**Figure 5 ijms-24-06950-f005:**
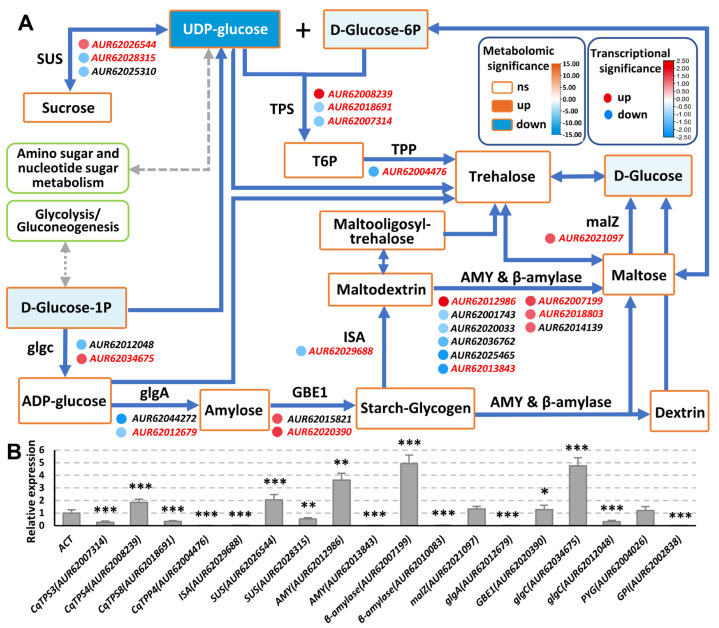
GO Analysis of starch and sucrose metabolism (main routes related to trehalose metabolism) in quinoa leaves. (**A**) Overview of the starch and sucrose metabolism pathway in quinoa leaves in response to saline-alkali stress. Rectangles with orange frames, metabolites; circles, DEGs; blue solid lines with arrows, directions of the processes; rectangles with green frames, other metabolism pathways; gray dotted lines, connection of the starch and sucrose metabolism pathway with other metabolism pathways. The colors of rectangles and circles indicate significances, which are presented in color scales. The genes highlighted in red were detected by RT-qPCR. (**B**) RT-qPCR analysis of the DEGs. *UBQ9* (AUR62020068) was used as endogenoue control. Three independent experiments per sample, three replicates per experiment. *, *p* < 0.05; **, *p* < 0.01; ***, *p* < 0.001; Student’s *t*-test. Abbreviations: AMY, α-amylase; GBE1, glycogen branching enzyme; glgA, glycogen synthase; glgC, ADPglucose pyrophosphorylase; ISA, isoamylase; malZ, maltodextrin glucosidase; SUS, sucrose synthase; T6P, trehalose 6-phosphate; TPP, trehalose 6-phosphate phosphatase; TPS, trehalose-6-phosphate synthase; UDP-glucose, uridine diphosphate glucose.

**Figure 6 ijms-24-06950-f006:**
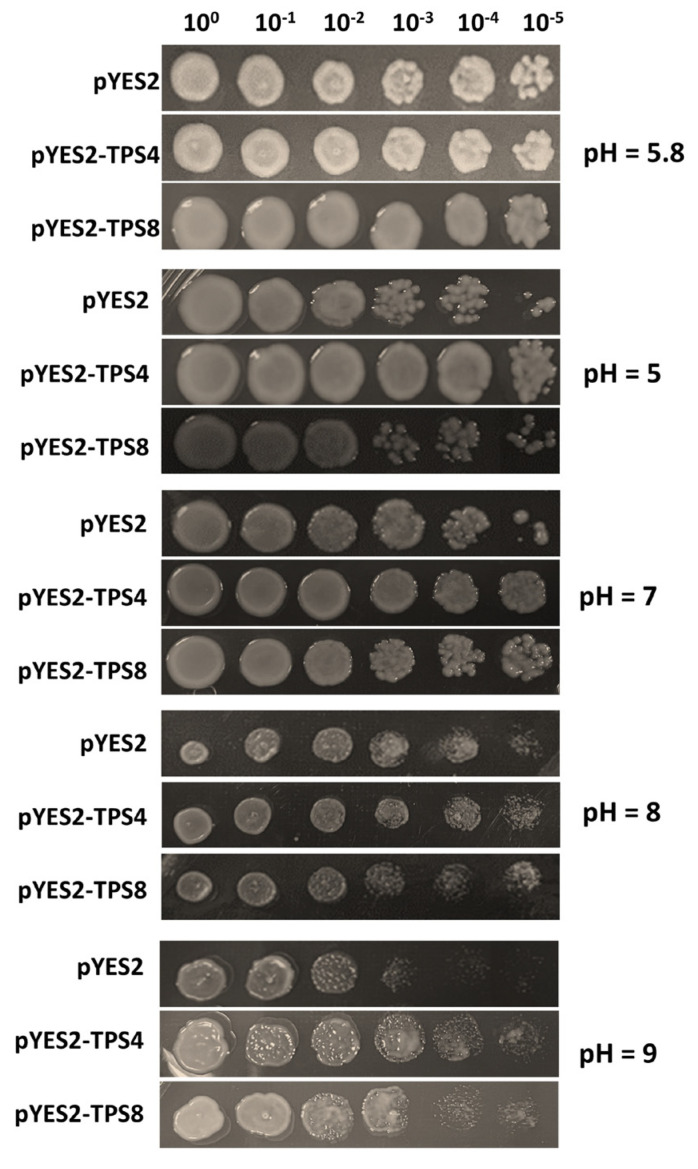
The growth activity of *CqTPS4-* and *CqTPS8*-expressing yeast under saline-alkali stress. Yeast strain INVSC1 was transformed with the *pYES2-CqTPS4, pYES2-CqTPS8,* or *pYES2* empty vector (control), respectively, and grew for 24 hrs on SC-Ura medium containing 25 mM NaHCO_3_ with pH value as indicated. pH 5.8 was the control pH value.

**Table 1 ijms-24-06950-t001:** Basic information of *CqTPS* genes in quinoa genome.

Gene Name	Locus_ID	Position (bp)			Deduced Polypeptide			Trehalose Ppase	
		Scaffold Location	Start	End	Length (aa)	MW (Da)	pI	Start (aa)	End (aa)
*CqTPS1*	AUR62002092	Scaffold_4480	2,006,754	2,013,924	857	96.63	6.24	595	829
*CqTPS2*	AUR62005352	Scaffold_1214	997,240	1,004,591	857	96.71	6.44	593	825
*CqTPS3*	AUR62007314	Scaffold_1971	3,942,433	3,949,016	863	97.47	6.70	597	832
*CqTPS4*	AUR62008239	Scaffold_3422	2,052,149	2,059,826	862	97.33	6.65	595	829
*CqTPS5*	AUR62013953	Scaffold_3035	858,836	875,850	919	103.22	6.48	616	818
*CqTPS6*	AUR62013957	Scaffold_3035	1,012,080	1,026,861	501	56.16	6.51	190	392
*CqTPS7*	AUR62014005	Scaffold_3035	2,206,377	2,210,382	830	93.91	6.54	634	798
*CqTPS8*	AUR62018691	Scaffold_1817	914,067	920,163	863	97.41	6.75	597	832
*CqTPS9*	AUR62019657	Scaffold_2127	7,449,003	7,470,166	963	108.42	6.63	651	854
*CqTPS10*	AUR62019658	Scaffold_2127	7,636,979	7,658,893	1032	116.20	6.20	729	931
*CqTPS11*	AUR62027638	Scaffold_1125	5,349,314	5,357,219	856	96.48	6.11	567	764
*CqTPS12*	AUR62035839	Scaffold_1759	1,363,165	1,381,138	1173	130.87	6.40	884	1087
*CqTPS13*	AUR62040284	Scaffold_1385	70,031	74,078	928	103.71	5.34	593	828

**Table 2 ijms-24-06950-t002:** Basic information of *CqTPP* genes in quinoa genome.

Gene Name	Locus_ID	Position (bp)			Deduced Polypeptide			Trehalose Ppase	
		Scaffold Location	Start	End	Length (aa)	MW (Da)	pI	Start (aa)	End (aa)
*CqTPP1*	AUR62002023	Scaffold_4480	1,360,051	1,362,652	356	40.67	10.05	100	333
*CqTPP2*	AUR62002194	Scaffold_4480	3,394,646	3,399,205	383	42.74	7.18	121	366
*CqTPP3*	AUR62003778	Scaffold_2370	4,555,540	4,557,764	356	40.63	9.98	100	333
*CqTPP4*	AUR62004476	Scaffold_4250	3,542,128	3,550,902	374	42.91	7.73	111	355
*CqTPP5*	AUR62006139	Scaffold_1001	2,668,322	2,671,355	346	39.33	7.41	86	328
*CqTPP6*	AUR62007476	Scaffold_1971	5,646,732	5,652,916	387	43.28	7.20	120	363
*CqTPP7*	AUR62015517	Scaffold_2751	7,485,997	7,489,524	383	42.84	7.21	121	366
*CqTPP8*	AUR62018916	Scaffold_3876	327,982	331,296	369	44.94	9.97	144	374
*CqTPP9*	AUR62023475	Scaffold_1606	414,991	423,431	324	37.08	5.97	61	306
*CqTPP10*	AUR62027730	Scaffold_3784	988,017	990,537	366	40.98	9.83	114	343
*CqTPP11*	AUR62039934	Scaffold_3651	310,550	317,771	386	43.03	6.80	120	363

## Data Availability

All data are presented in this article.

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
