# Peer review of "Genome-Wide Identification and Analysis of Stress Response of Trehalose-6-Phosphate Synthase and Trehalose-6-Phosphate Phosphatase Genes in Quinoa"

_ijms, 2023, doi:10.3390/ijms24086950_

Round 1

Reviewer 1 Report

Dear Editors,

Thank you so much for choosing me as a reviewer of the manuscript
ijms-2294215  Genome-wide Identification and analysis of stress response of Trehalose-6-Phosphate Synthase and Trehalose-6-Phosphate Phosphatase Genes in quinoa” submitted to International Journal of Molecular Sciences. I hope that my manuscript will help Authors to improve their manuscript.

Detailed remarks concerning manuscript.

Abstract. Not only the clear aim of the report and scientific hypothesis should be stated together with the answer to question stated as scientific hypothesis. The abstract should include the background of the studies, aim of the report, methodology, results and one or two general concluding sentences. Therefore I suggest doing needed changes. Please include the one or two methodology sentences. Key words. It is not recommended to use as key words the words or phrases used in the title of the manuscript. Please do needed changes. Please correct “Rehalose-6-phosphase synthase (TPS)’’ where needed.

All figures and tables should be clear for the reader without referring to the text of the manuscript. Please add the information or explanations where needed.

Conclusions: Please provide the specific practical application for the results of the studies as well as the directions for the future studies.

In the methodology section The Latin name of the quinoa should be provided.

Please provide the information concerning practical application of the presented studies.

 References. There are many editorial mistakes. It is impossible to mention all of them. There are some examples: Once the abbreviated but the other time full journal names are provided. Once only the first word of the manuscript title is written with capital letter, but the other time each word of the manuscript title is written with capital letter. The reference list should prepared strictly according to the guidelines for authors. Please go very carefully through the whole reference list and do needed changes.

Author Response

Thanks very much for taking your time to review this manuscript. I really appreciate all your comments and suggestions! The corrected sections have been marked in red in revised manuscript. A point-by-point response to the comments of 2 reviewers is followed as below.

Responses to Reviewers' comments:

Reviewer #1:

Abstract. Not only the clear aim of the report and scientific hypothesis should be stated together with the answer to question stated as scientific hypothesis. The abstract should include the background of the studies, aim of the report, methodology, results and one or two general concluding sentences. Therefore, I suggest doing needed changes. Please include the one or two methodology sentences.

Response: Thanks for your suggestion. We have added methodology description.

Key words. It is not recommended to use as key words the words or phrases used in the title of the manuscript. Please do needed changes. Please correct “Rehalose-6-phosphase synthase (TPS)’’ where needed.

Response: Thank you for your suggestion. We have changed the key words.

Figures and tables: All figures and tables should be clear for the reader without referring to the text of the manuscript. Please add the information or explanations where needed.

Response: Thank you for your helpful advice. We have added the information and explanations needed.

Conclusions: Please provide the specific practical application for the results of the studies as well as the directions for the future studies.

Response: Thank you for your suggestion. We have added specific practical application for the results of the studies as well as the directions for the future studies in Conclusions.

Methodology: In the methodology section The Latin name of the quinoa should be provided.

Response: Thank you for your suggestion. We have added the Latin name of the quinoa.

References: Please provide the information concerning practical application of the presented studies.

References. There are many editorial mistakes. It is impossible to mention all of them. There are some examples: Once the abbreviated but the other time full journal names are provided. Once only the first word of the manuscript title is written with capital letter, but the other time each word of the manuscript title is written with capital letter. The reference list should prepare strictly according to the guidelines for authors. Please go very carefully through the whole reference list and do needed changes.

Response: Thank you for pointing out the problem. We had revised the mistakes in References.

Reviewer 2 Report

The paper is focused on the identification and analysis of the stress response gene in Quinoa. Generally, the article is exciting and could be published. There are a few suggestions:

The introduction could be shortened.

In the discussion part, please add more about transcriptome analysis.

Author Response

Reviewer #2:

Thanks very much for taking your time to review this manuscript. I really appreciate all your comments and suggestions! The corrected sections have been marked in red in revised manuscript. A point-by-point response to the comments of 2 reviewers is followed as below.

Introduction: The introduction could be shortened.

Response: Thanks for your suggestion. We have revised the introduction to make it shorter.

Discussion: In the discussion part, please add more about transcriptome analysis.

Response: Thank you for your kind suggestion. We have added the transcriptome analysis about the Starch and sucrose metabolic pathways.
